# The development of children born to young mothers with no, first- or second-generation HIV acquisition in the Eastern Cape province, South Africa: a cross-sectional study

Lorraine Sherr [1], Katharina Haag [2], Kathryn J Steventon Roberts,[1,3] Lucie Dale Cluver,[3,4] Camille Wittesaele,[3,5] Bongiwe Saliwe,[6] Janke Tolmay,[6] Nontokozo Langwenya,[3] Janina Jochim,[3] Wylene Saal,[7] Siyanai Zhou,[6] Marguerite Marlow,[8] Jenny J Chen-Charles [3], Elona Toska [3,6,9]

For numbered affiliations see end of article.

**Correspondence to**
Professor Lorraine Sherr;
l.sherr@ucl.ac.uk

## ABSTRACT

**Background** The intergenerational effects of HIV require long-term investigation. We compared developmental outcomes of different generations impacted by HIV—children of mothers not living with HIV, the 'second generation' (ie, with recently infected mothers) and the 'third generation' (ie, children of perinatally infected mothers).

**Methods** A cross-sectional community sample of N=1015 young mothers (12–25 years) and their first children (2–68 months, 48.2% female), from South Africa's Eastern Cape Province. 71.3% (n=724) of children were born to mothers not living with HIV; 2.7% (n=27; 1 living with HIV) were third-generation and 26.0% (n=264; 11 living with HIV) second-generation children. Child scores on the Mullen Scales of Early Learning (MSEL), the WHO Ten Questions Screen for Disability and maternal demographics were compared between groups using $\chi^2$ tests and univariate approach, analysis of variance analysis. Hierarchical linear regressions investigated predictive effects of familial HIV infection patterns on child MSEL composite scores, controlling for demographic and family environment variables.

**Results** Second-generation children performed poorer on gross (M=47.0, SD=13.1) and fine motor functioning (M=41.4, SD=15.2) and the MSEL composite score (M=90.6, SD=23.0) than children with non-infected mothers (gross motor: M=50.4, SD=12.3; fine motor: M=44.4, SD=14.1; composite score: M=94.1, SD=20.7). The third generation performed at similar levels to non-exposed children (gross motor: M=52.4, SD=16.1; fine motor: M=44.3, SD=16.1, composite score: M=94.7, SD=22.2), though analyses were underpowered for definite conclusions. Hierarchical regression analyses suggest marginal predictive effects of being second-generation child compared with having a mother not living with HIV (B=−3.3, 95% CI=−6.8 to 0 .1) on MSEL total scores, and non-significant predictive effects of being a third-generation child (B=1.1, 5% CI=−7.5 to 9.7) when controlling for covariates. No group differences were found for disability rates (26.9% third generation,

27.7% second generation, 26.2% non-exposed; $\chi^2$=0.02, p=0.90).

**Conclusion** Recently infected mothers and their children may struggle due to the disruptiveness of new HIV diagnoses and incomplete access to care/support during pregnancy and early motherhood. Long-standing familial HIV infection may facilitate care pathways and coping, explaining similar cognitive development among not exposed and third-generation children. Targeted intervention and fast-tracking into services may improve maternal mental health and socioeconomic support.

## STRENGTHS AND LIMITATIONS OF THIS STUDY

⇒ To our knowledge, this is the first study to investigate developmental outcomes and life circumstances of the 'third generation', that is, children with a perinatally infected mothers.

⇒ A particular strength is the comparison to both children of the 'second generation' (ie, with recently infected mothers) and children not affected by HIV but living in similar high-risk environments characterised by poverty and deprivation.

⇒ While we found that children of the third generation perform equally well or better than children not directly exposed to HIV, the sample of the former was small, limiting our power to draw a definite conclusion.

⇒ Our sample consisted predominantly of young mothers. While both rates of HIV infection and young pregnancy may be compounded in deprived areas, the current findings need to be replicated in a more representative samples.

## INTRODUCTION

Approximately 7.5 million people in South Africa were living with HIV in 2020, including 360 000 adolescents aged 10–19 years.[1] Generally, HIV infections have been

linked to higher rates of morbidity, stigma, poverty and other adverse outcomes.[2 3] While rates of perinatal HIV infection are declining, due to improved screening and treatment options,[4–6] historical exposure means that an increasing number of perinatally infected children are now reaching adolescence and young adulthood and are becoming parents themselves. There is a need to study their children's developmental outcomes, in order to devise effective support for these families.

To the knowledge of the authors, to date, there have been limited studies focusing on this 'third generation', that is, children who have their mother perinatally infected by HIV. Most designs do not differentiate between mothers who were recently or perinatally infected and instead focus on comparing HIV infected, HIV exposed and uninfected and HIV unexposed children.[7–9] Findings suggest that the former two groups may show developmental sequelae (including cognitive and motor), which can be explained by exposure to the virus (in utero) and Antiretroviral therapies (ART) as well as environmental factors associated with HIV infection (eg, prematurity, poor parental mental health and substance use, stigma, caretaker changes, food insecurity).[10 11] However, specific knowledge on the third generation is still extremely limited. The few existing studies focus mainly on maternal characteristics (eg, ART adherence, age at pregnancy) and child infection status.[12]

Of note, previous research on adolescents who were perinatally infected (potential mothers of the third generation) vs recently infected (potential mothers of the second generation) indicates that they may be facing different types of life challenges that could affect their parenting in divergent ways. Those perinatally infected tend to have experienced more chronic adversity, such as parental and/or sibling illness and death, caretaking responsibilities, poorer parenting and compromised parent–child relationships, poor wider family and community support, and compounded internalised and associative stigma, all of which may increase the risk for lower well-being and higher substance use.[2 4 13–17] Those recently infected have more likely been marginalised and struggle with the impact of their diagnosis, including biographical disruptions and reduced hope for the future, and to have difficulties with accessing and engaging in care. All of these issues may be compounded by mental health problems including anxiety, depression, substance use and internalised stigma.[14 17–19]

The current study was set up to explore the development of young children (2–68 months) of adolescent and young adult mothers living in the Eastern Cape of South Africa, with a specific aim to gain a better understanding of third- and second-generation HIV effects. HIV infections often tend to cluster in disadvantaged environments, leading to disproportionally affected communities, where risks of initial HIV infection and HIV being passed on across generations are heightened. These communities are often also affected by high adolescent pregnancy rates,[20] which are associated with a range of risks for mother and child such as poor educational prospects, exposure to poverty and violence, and poor developmental outcomes.[21 22] Furthermore, they can also be a marker for adolescent sexual risk behaviours, and there may be a syndemic occurrence between HIV and adolescent parenthood risk among young women in sub-Saharan Africa.[22]

We had two main analysis aims: (1) to compare child developmental scores (fine and gross motor, visual reception, expressive and receptive language skills) at ages 2–68 months, independent of child HIV status, in children of mothers 19 years or younger who were not living with HIV, were recently infected through sexual exposure (second generation) and were perinatally infected (third generation) and (2) to investigate how mothers of these groups may differ in terms of their socioeconomic status, well-being and social support.

## METHODS

### Participants

Data used within these analyses originate from the 'Helping Empower Youth Brought up in Adversity with their Babies and Young children' study. The study was conducted in rural and periurban health districts of South Africa's Eastern Cape and aimed at investigating the effects of adolescent motherhood, as well as intergenerational effects of HIV exposure. A total of 1046 adolescent and young adult mothers (10–25 years) with at least one living child were interviewed between March 2018 and July 2019.[23] The required sample size was estimated based on expected effect sizes for key outcomes. Participants were partially recruited from a previous study (n=159: any young mothers included), as well as through six parallel sampling strategies developed in cooperation with local experts and an adolescent mother advisory group to ensure representativeness (n=887: only adolescent mothers). This comprised sampling through 73 known health facilities within the districts, 43 secondary schools, 9 maternity units and referrals by service providers, social workers and adolescent mothers themselves. A total of 95%–98% of eligible mothers from each recruitment channel were successfully enrolled into the study. The current analyses were limited to data on first child of the young mother only (10–25 years), and children aged 68 months or younger who were within the normed range for the Mullen Scales (n=31 excluded based on this age restriction), limiting the sample to n=1015. We also performed analyses including adolescent mothers (age at pregnancy ≤19, n=972) only for the key outcomes (see online supplemental appendix 1), as these mothers had been the main recruitment focus for the study.

### Patient and public involvement

The study team were advised on recruitment methods by adolescent mothers, whose suggestions were included in the study protocol. Furthermore, the team has been working with two Teen Advisory groups in the Eastern

and Western Cape of South Africa, who were involved in piloting the study. Feedback was incorporated to improve relevance and acceptability of the questionnaire and the research procedure. Finally, through the engagement of community leaders, it was ensured that recruitment strategies were effective, sensitive to participant's circumstances and minimised stigma risks.

## Procedure

The dataset analysed combined four data sources. First, all adolescent mother participants completed a detailed study questionnaire relating to sociodemographic characteristics, sexual and reproductive health, physical and mental health, relationships and social support. Second, they completed an adolescent parent questionnaire, which collected data on maternal and child health, child development, the father of the child and maternal factors including social support, the parenting experience and violence exposure. Third, cognitive assessments of the children were performed by a trained administrator, using the Mullen Scales of Early Learning. Finally, details from the child's medical records (Road to Health Booklet) were also included in the database. All participants provided written consent, and interviews were conducted in the language of their choice (ie, English or isiXhosa). Regardless of participation, all adolescents received a small 'snack pack' containing a snack and juice during the interview, and a small 'thank you pack', with personal products such as toothpaste and a toothbrush. Items included in these packs were selected by the adolescent advisory group as preferable and appropriate. New mothers also received a 'baby pack', the contents of which were also chosen by an adolescent advisory group, and included nappies and baby cream.

## Measures
### Mode of maternal HIV acquisition
Mode of maternal HIV acquisition (perinatal vs recent) was assessed through an algorithm, given that the study was community-based and not linked to clinical testing data. Accordingly, it was derived using a logic tree based on clinical and fieldwork experiences .[17] The algorithm allowed for categorisation according to self-report, age of ART initiation, and parental death information (Tolmay, Saal et al., in preparation). For the current analyses, we compared the following three groups: children of the third generation (mother perinatally infected), children of the second generation (mother recently infected) and children of mothers not living with HIV (see figure 1). Child HIV status was not taken into account in these classifications, since absolute numbers were low (n=12 based on maternal self-reported data), with only one of these children being in the third generation. Thus, we were underpowered to study these children separately. We decided however to retain them in the analyses since they are relevant members of the second and third generations, respectively.

### Child developmental outcomes
Child developmental outcomes were assessed using the Mullen Scales of Early Development (MSEL).[24] The MSEL is normed for children aged 0–68 months (USA) and assesses child performance across five domains: gross motor (only for age <39 months), fine motor, visual reception, receptive and expressive language (score range: 20–80). A composite score (score range: 49–155) can be derived, and the scales have been validated for use in sub-Saharan Africa.[23 25 26] Child disability status was assessed using the WHO Ten Questions Screen for Disability.[27] This measure can be applied to detect common disabilities (hearing, visual, physical, speech, mental and epilepsy) in children. A score indicating any disability ('yes' across any of the 10 items) was derived for the current analyses.

### Maternal variables
Adolescent mothers were compared on various variables, including demographic factors (maternal age, maternal and paternal age at pregnancy), child feeding method used (formula, breastfeeding, combined, other), maternal school progression (self-report: repeated at least one school grade), poverty (number of the eight socially perceived necessities for children the family had access to,[28] household access to any government cash transfers child support, foster child, pension, disability or care dependency grant, measured via South Africa Census item,[29] food security number of days there was not enough food for the household in the past 7 days,[30] full ART adherence over the past 7 days (Patient Medication Adherence Questionnaire,[31] any HIV clinic appointments missed in the past year, extent of HIV-related stigma Adolescents Living with HIV Stigma Scale,[32] depressive symptoms in the past 2 weeks Child Depression Inventory-Short Form,[33] anxiety symptoms in the past month Revised Children's Manifest Anxiety Scale,[34] PTSD symptoms in the past month Child PTSD Checklist-Short Form,[35] suicidality symptoms during the past month Mini International Psychiatric Interview for Children and Adolescents- Suicidality and Self-Harm Subscale,[36] extent of community violence exposure in the past year item from the Child Exposure to Community Violence Checklist,[37] intimate partner violence exposure in the past month (Juvenile Victimisation Questionnaire,[38] current parenting stress (Parental Stress Scale[39]) and amount of social support received (adjusted version of the Medical Outcomes Study Social Support Survey,[40] previously used in the South African context.[41]

### Statistical analyses
All analyses were conducted using STATA V.16 SE. First, child and adolescent mother descriptive information for the three groups of interest (children of the third generation, children of the second generation and HIV-unexposed children) was provided. Next, child developmental outcomes and maternal factors surrounding socioeconomic status, HIV-related variables (only HIV-affected mothers), mental health and social environment

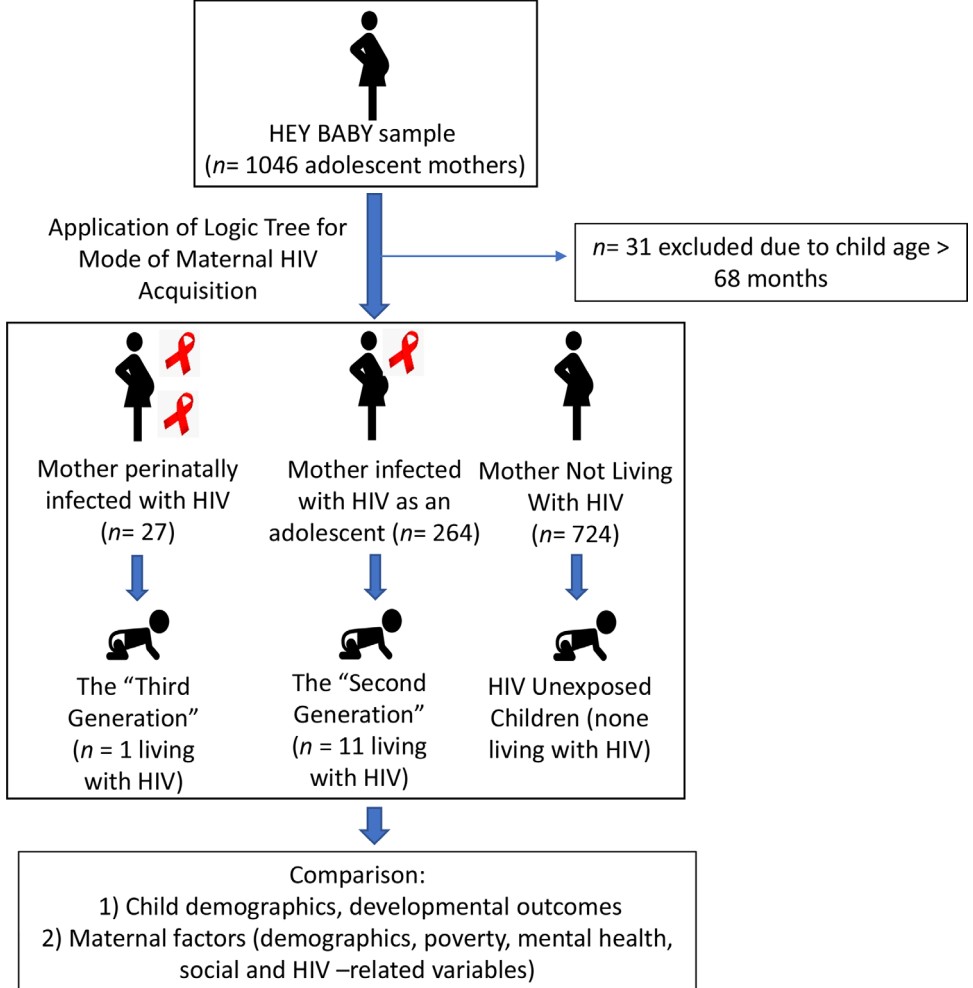

HEY BABY sample
(*n*= 1046 adolescent mothers)

Application of Logic Tree for Mode of Maternal HIV Acquisition

*n*= 31 excluded due to child age > 68 months

Mother perinatally infected with HIV
(*n*= 27)

Mother infected with HIV as an adolescent (*n*= 264)

Mother Not Living With HIV
(*n*= 724)

The "Third Generation"
(*n* = 1 living with HIV)

The "Second Generation"
(*n* = 11 living with HIV)

HIV Unexposed Children (none living with HIV)

Comparison:
1) Child demographics, developmental outcomes
2) Maternal factors (demographics, poverty, mental health, social and HIV –related variables)

**Figure 1** Flow chart of classification by mode of maternal HIV acquisition. HEY BABY, Helping Empower Youth Brought up in Adversity with their Babies and Young children.

were compared between groups, using $\chi^2$ tests and univariate analyses of variance as appropriate. For continuous variables, Tukey's range tests were used to conduct post hoc group comparisons. Effects at p<0.05 were considered relevant. Finally, a hierarchical regression model was run, with the child composite score on the Mullen scales as a key outcome. In a first step, child familial HIV exposure (third generation, second generation, HIV-unexposed) was added as a predictor. In a second step, we included relevant control variables that were found to be associated with the predictor or outcome variables in univariate analyses or were identified as relevant within the literature to see whether child familial HIV exposure would remain a relevant predictor.

## RESULTS
### Descriptive variables
After applying the logic tree for mode of maternal HIV acquisition, 27 mothers (2.7%) were classified as perinatally infected, 264 (26.0%) as recently/sexually infected, and 724 (71.3%) as not living with HIV. Table 1 presents adolescent mother and child sample characteristics, split

according to whether the child was in the third generation, the second generation or not exposed to HIV. There were no significant group differences in child sex, HIV status and crèche attendance; however, children of the second generation were on average older than those without HIV exposure (diff=6.7, 95% CI=4.3 to 9.1). No significant age differences were found between children of the third generation and the other two groups (diff=2.6, 95% CI=−4.2 to 9.5). Formula (without breast feeding) use was the most common feeding method during the first 6 months of the child's life for mothers of the third generation (63.0%). For mothers of the second generation, formula use (only) and breast feeding (only) were similarly common (35.0% and 38.4%), while the majority of mothers of HIV-unexposed children opted for a mix of both (40.9%).

Similar to their children, mothers of the second generation (M=19.7 years) were slightly older than those of third generation (M=18.7 years), and the latter were older than mothers of HIV-unexposed children (M=17.7 years) (F=148.01, p<0.001). Furthermore, mothers of the second and third generation were older at pregnancy with the

**Table 1** Sociodemographic characteristics of adolescent mothers and their children according to familial patterns of HIV infection

| | Overall (N=1015) | Third generation (n=27) | Second generation (n=264) | No HIV exposure (n=724) | P value |
|---|---|---|---|---|---|
| **Child variables** | | | | | |
| Age in months (M, SD) | 18.7 (14.7) | 20.9 (16.7) | 23.5 (16.4)* | 16.8 (13.5) | <0.001 |
| Sex (1=female) | 489 (48.2%) | 16 (59.3%) | 123 (48.2%) | 350 (48.3%) | 0.46 |
| HIV status | 12 (5.0%) | 1 (9.1%) | 11 (10.0%) | – | 0.92 |
| **Feeding** | | | | | |
| Breast feeding and formula | 347 (34.2%) | 4 (14.8%) | 47 (17.9%) | 296 (40.9%) | <0.001 |
| Baby formula only | 318 (31.4%) | 17 (63.0%) | 92 (35.0%) | 209 (28.9%) | |
| Breastfeeding only | 258 (25.4%) | 5 (18.5%) | 101 (38.4%) | 152 (21.0%) | |
| Other milk or water | 91 (9.0%) | 1 (3.7%) | 23 (8.8%) | 67 (9.3%) | |
| Crèche attendance | 304 (29.8%) | 8 (29.6%) | 80 (30.4%) | 216 (29.8%) | 0.98 |
| **Maternal variables** | | | | | |
| Age in years (M, SD) | 18.2 (1.8) | 18.7 (1.6)† * | 19.7 (2.0)* | 17.7 (1.5) | <0.001 |
| Age at pregnancy with the participating child (years; M, SD) | 16.8 (1.7) | 17.3 (2.0) | 17.8 (1.9) | 16.3 (1.4) | <0.001 |
| Paternal age at pregnancy (years, M, SD) | 21.1 (3. 9) | 21.3 (4.3)† | 23.2 (4.08)* | 20.4 (3.5) | <0.001† |
| Informal housing | 217 (21.9%) | 6 (23.1%) | 60 (23.6%) | 151 (21.2%) | 0.76 |
| Living in rural area | 293 (28.9%) | 8 (29.6%) | 63 (23.9%) | 222 (30.7%) | 0.11 |
| Any school grade repeated | 581 (60.8%) | 16 (61.5%) | 169 (69.3%) | 396 (57.8%) | 0.007 |
| Current relationship | 670 (66.5%) | 21 (77.8%) | 186 (71.8%) | 463 (64.2%) | 0.038 |
| **Poverty and social protection** | | | | | |
| No of eight necessities available | 5.2 (2.2) | 4.4 (2.7) | 4.8 (2.3)* | 5.4 (2.1) | <0.001 |
| Any household grant | 944 (92.5%) | 26 (96.3%) | 254 (96.2%) | 659 (91.1%) | 0.017 |
| Days insufficient food for child (past week) | 0.3 (1.1) | .0 (.2) | 0.4 (1.2) | 0.3 (1.0) | 0.07 |

Group comparisons were conducted using $\chi^2$ tests for categorical variables and univariate analyses of variance for continuous variables. Tukey's range post hoc tests were undertaken to identify mean differences between groups for continuous variables.
*Statistically different from HIV unexposed group (p<0.05). Ns vary depending on missing values (N=933–1015).
†Statistically different from the second generation (p<0.05).

participating child (M=17.8 and M=17.3 vs M=16.3 years, respectively; F=89.23, p<0.001) compared with mothers of HIV-unexposed children. Fathers of the second generation (M=23.2 years) were on average older at childbirth than those of the third generation and HIV-unexposed children (M=21.3 and M=20.4 years respectively, F=49.08, p<0.001). There was no difference between groups regarding whether the mother lived in informal housing or a rural area, though mothers unaffected by HIV were marginally less likely to be in a relationship (64.3%, vs 77.8%% for the third generation and 71.5% for the second generation; $\chi^2$=6.52; p=0.038). Mothers of the second generation were more likely to have repeated a school grade (69.3%), vs 61.5% for mothers of the third generation and 57.9% for mothers of HIV-unexposed children ($\chi^2$=9.91, p=0.007).

Mothers of the second and third generations had on average fewer of the eight necessities available (F=10.11, p<0.001) (see table 1); post hoc tests identified significant differences for second generation (p<0.001) and non-significant differences for third generation mothers (p=0.060) compared with those not living with HIV. Mothers of the second and third generations were also more likely to receive household cash grants ($\chi^2$=8.10;

p=0.017). However, mothers of the third generation had the lowest average days on which they had no food available for the child in the past week, followed by mothers not living with HIV, and then recently infected mothers, though the overall model was only close to significant (F=2.73; p=0.066).

### Child developmental outcomes

Table 2 presents child developmental outcomes according to familial patterns of HIV infection. For the Mullen Scales of Early Child Development, we found that the third generation had the highest gross motor scores (M=52.4, SD=16.1), ahead of children not exposed to HIV (M=50.4, SD=12.3) and the second generation (M=47.0, SD=13.1; F=6.50, p=0.001). However, due to low power resulting from the small sample of children in the third generation, Tukey's range post hoc tests only identified a difference between the latter two groups (diff=3.41, 95% CI=1.08 to 5.73). Children of the second generation (M=41.4, SD=15.2) exhibited poorer fine motor scores than both HIV non-exposed children (M=44.4, SD=14.1) and children of the third generation (M=44.3, SD=16.1; F=4.26, p=0.014), though post hoc tests again only indicated differences between the former two groups

**Table 2** Infant (0–68 months) developmental outcomes, according to familial patterns of HIV infection

| | Overall(N=1015) | Third generation (n=27) | Second generation (n=264) | No HIV exposure (n=724) | P value |
|---|---|---|---|---|---|
| Mullen Scales of Child Development (M, SD) | | | | | |
| Gross Motor Score† | 49.6 (12.7) | 52.4 (16.1) | 47.0 (13.1)* | 50.4 (12.3) | 0.002 |
| Visual Reception Score | 42.1 (14.2) | 43.9 (14.7) | 40.7 (14.3) | 42.6 (14.2) | 0.14 |
| Fine Motor Score | 43.6 (14.6) | 44.3 (16.1) | 41.4 (15.2)* | 44.4 (14.2) | 0.014 |
| Receptive Language Score | 47.4 (13.6) | 47.7 (13.8) | 46.1 (15.0) | 47.8 (13.0) | 0.21 |
| Expressive Language Score | 51.6 (13.5) | 51.7 (13.4) | 50.7 (14.8) | 51.9 (13.0) | 0.48 |
| Composite Score | 93.2 (21.4) | 94.6 (23.4) | 90.6 (23.0)* | 94.1 (20.7) | 0.066 |
| WHO Disability Score-Any Disability (N, %) | 242 (26.7) | 7 (26.9) | 69 (27.7) | 166 (26.2) | 0.90 |
| Any Days Sick (past month) (N, %) | 218 (21.8) | 4 (17.4) | 41 (15.6) | 173 (24.2) | 0.013 |

Group comparisons were conducted using $\chi^2$ tests for categorical variables and univariate analyses of variance for continuous variables.
*Statistically different from HIV unexposed group (p<0.05)
†Gross motor score: n=831. Mullen subdomain scores range from 20 to 80, composite scores from 49 to 155. Tukey's range post hoc tests were undertaken to identify mean differences between groups for continuous variable
‡Statistically different from the second generation (p<0.05)

(diff=3.04, 95% CI=0.59 to 5.49). No differences were found on the visual reception, receptive and expressive language subscales, though scores for the second generation were consistently lowest, while children of the third generation tended to do as well or even better than non-exposed children. In terms of the composite scores, children of the second generation performed marginally lower (M=90.6, SD=23.0) than those not exposed to HIV (M=94.1, SD=20.7; p=0.055) in post hoc tests (diff=3.55, 95% CI=−0.05 to 7.16). Differences of the former to third generation (M=94.7, SD=22.2) were not significant in post hoc tests (p=0.66), despite the scores of the third generation being approximately equal to those of non-infected children (overall model: F=2.72 p=0.066). Of note, all groups performed lower than expected compared with the US reference population based on which the Mullen Scales were developed (M=100.0 for the composite score; p<0.001). No group differences were found for presence of any disability ($\chi^2$=0.02, p=0.90). Finally, the rate of children who had been sick at least 1 day during the past month was lowest for the second generation (15.6%), with increasing rates towards the third generation (17.4%) and those not exposed to HIV (24.2%) ($\chi^2$=8.62, p=0.013). When limiting the sample to adolescent mothers (age ≤19) only, the same magnitude and directions of findings were obtained (see online supplemental appendix 1).

### Maternal characteristics for the three subgroups
Mothers of the second and third generations did not differ regarding ART adherence, clinic appointments missed, and HIV-related stigma (p=0.69-.95). Both groups were about half as likely to have been in excellent physical health in the past twelve months as mothers of non-exposed children ($\chi^2$=108.46; p<0.001). Mental health symptom scores were overall low, and no differences were found between groups in terms of anxiety (F=0.89, p=0.41) and only marginal differences for PTSD, with mothers from the second generation scoring higher than bother other groups (F=2.38 p = 0.093). Mothers of the second generation also had the highest depression scores (F=6.12, p=0.002), with post hoc tests indicating significant differences to those not HIV exposed (p=0.005) and marginal differences to mothers of third generation (p=0.053). Second generation mothers also scored substantially higher than both other groups on suicidality levels (F=4.11, p=0.017), though due to low power, only significant differences to mothers not living with HIV were identified in post hoc tests (p=0.017). Finally, no group differences were detected for exposure to intimate partner and community violence or parenting stress (p=0.39-.97). However, mothers of the third generation indicated the lowest levels of social support, followed by the second generation and then mothers of non-exposed children (F=3.98, p=0.019), though post hoc tests indicated only close to significant differences between mothers of the third generation and those of HIV-unexposed children (p=0.061; see table 3).

### Regression models for predicting child cognitive development composite score
Table 4 shows the results of the hierarchical linear regression model predicting the child composite score on the Mullen Scales. In step 1, compared with the baseline class of children not exposed to HIV, being a child of the second generation was associated with a lower developmental score (p=0.021); no predictive effect was found for being a child of the third generation (p=0.91; model: F(2,1012)=2.72, p=0.066). The former association was retained at a marginal level in the adjusted model in step 2 (p=0.059; overall model: F(8,889) = 4.08, p<0.001), with mothers having repeated a school grade additionally being predictive of poorer outcomes, and higher maternal age at birth exerting a protective influence. No predictive effects were found for maternal depression, social support, number of necessities and food insecurity. Overall, however, the amount of variance explained by both models was relatively low.

### DISCUSSION
In this sample of young mothers and their infants living in South Africa, we found that children of the second

**Table 3** Comparison of maternal life circumstances, according to familial patterns of HIV infection

| | Overall (N=1015) | Third generation (n=27) | Second generation (n=264) | No HIV exposure (n=724) | P value |
|---|---|---|---|---|---|
| **HIV-associated variables** | | | | | |
| Full ART adherence past 7 days | 213 (75.8%) | 19 (79.2%) | 194 (75.5%) | – | 0.69 |
| Any clinic appointments missed past month | 10 (4.1%) | 1 (4.6%) | 9 (4.1%) | – | 0.92 |
| Any stigma | 56 (20.3%) | 5 (20.8%) | 51 (20.2%) | – | 0.95 |
| **Physical and mental health** | | | | | |
| Excellent physical health past 12 months | 598 (58.6%) | 9 (33.3%) | 88 (33.3%) | 499 (68.9%) | <0.001 |
| Depression Score past 2 weeks (0–20) | 0.7 (1.4) | 0.3 (.7) | 1.0 (1.7)* | 0.6 (1.3) | 0.002 |
| Anxiety Score past month (0–14) | 0.8 (1.9) | 0.7 (1.6) | 0.6 (1.9) | 0.8 (1.9) | 0.41 |
| PTSD Score past month (0–36) | 3.0 (5.6) | 2.0 (4.9) | 3.6 (5.6) | 2.9 (4.9) | 0.09 |
| Suicidality Score past month (0–5) | 0.2 (.8) | 0.1 (.3) | 0.3 (1.1)* | 0.1 (.7) | 0.017 |
| **Social and environmental factors** | | | | | |
| Parenting Stress Score (18–90) | 24.8 (5.8) | 24.7 (4.8) | 24.9 (5.6) | 24.8 (5.8) | 0.97 |
| Social Support Score (0–14) | 13.4 (2.04) | 12.7 (2.5) | 13.2 (2.3) | 13.5 (1.9) | 0.019 |
| Extent of Intimate Partner Violence Exposure (0–4) | 0.1 (.6) | 0.1 (.4) | 0.2 (.7) | 0.1 (.5) | 0.39 |
| Extent of Community Violence Exposure (0–4) | 0.3 (.8) | 0.2 (.6) | 0.3 (.8) | 0.3 (.8) | 0.79 |

Group comparisons were conducted using $\chi^2$ tests for categorical variables and univariate analysis of variance for continuous variables. Tukey's range post hoc tests were undertaken to identify mean differences between groups for continuous variable.
Ns vary depending on missing values (N=950–1015; N=828 for Intimate Partner Violence—low response rates may be due to the sensitive topic).
*Statistically different from HIV unexposed group (p<0.05).
†Statistically different from the second generation (p<0.05).
ART, antiretroviral therapy.

generation (ie, with mothers recently infected with HIV) scored significantly lower on gross motor and fine motor functioning. For the overall composite score they tended to show lower scores as well than children not exposed to HIV. Children of the third generation (ie, of perinatally infected mothers) appeared to score similarly or even higher than children not exposed to HIV across all domains. However, the comparatively small sample size in this group makes it challenging to draw definite conclusions. Mothers also differed in their life circumstances, with mothers of the second generation showing higher rates of depression and suicidality symptoms, as well as lower food security, while mothers of the third generation indicated lower social support. No differences were found on adherence and clinical attendance variables.

**Table 4** Multiple regression analysis predicting child developmental score from familial patterns of HIV infection in step 1, with step 2 controlling for relevant child and maternal variables

| | Unstandardised regression weights (B; 95% CI) | Standardised regression weights (β) | P value |
|---|---|---|---|
| **Step 1: unadjusted model** | | | |
| Mother not living with HIV (Ref.) | – | | |
| Mother perinatally infected | .46 (–7.77 to 8.69) | 0.00 | 0.91 |
| Mother recently infected | −3.5 (–6.57 to −0.53) | −0.07 | 0.021 |
| Adjusted R² | 0.003 | | |
| **Step 2: adjusted model** | | | |
| Mother not living with HIV (Ref.) | – | | |
| Mother perinatally infected | 1.06 (–7.51 to 9.65) | 0.01 | 0.81 |
| Mother recently infected | −3.36 (–6.85 to .12) | −0.07 | 0.059 |
| Maternal Depression Score | −0.45 (–1.46 to .57) | −0.03 | 0.39 |
| Maternal Social Support | .21 (–0.47 to .89) | 0.02 | 0.54 |
| No of Necessities | .36 (–0.30 to .1.03) | 0.04 | 0.29 |
| Maternal age at pregnancy | 1.10 (.21 to 1.99) | 0.09 | 0.016 |
| Mother repeated any school grade | −5.98 (–8.87 to −3.07) | −0.14 | <0.001 |
| No of days insufficient food for child | −0.32 (–1.66 to 1.01) | −0.02 | 0.64 |
| Adjusted R² | 0.027 | | |

Step 2 (adjusted model) based on N=898, due to missing data on the control variables. B=unstandardised regression weights. β=standardised regression weights.

Any interpretation of the current data warrants caution, due to low power resulting from the small sample size of the third-generation group. Still, our data appear to provide a tentative first indication that children of the third generation could be doing similarly well as those not exposed to familial HIV across most developmental domains. This would be promising and may indicate that families affected by HIV are in the long-term profiting from efforts to provide social protection, links to services and antiretroviral treatment, which could prevent chronic adversity from being perpetuated. However, it remains to be seen whether such benefits are sustained, with studies in children living with HIV suggesting positive early developmental outcomes, but declines with age.[26 42] While the aim is to ultimately prevent perinatal HIV infections, larger studies of already infected young adults and their children may provide further insights into the development and needs of children of the third generation.

Children of the second generation, on the other hand, tended to do poorer across several domains, an effect that was marginally retained when controlling for maternal mental health, social support and sociodemographic factors. The shock of maternal recent HIV diagnosis may have placed a major burden on these families, compounded by risk drivers which may have accounted for the HIV exposure in the first place, such as economic and food insecurity, or mental health struggles. This indicates that affected families may profit from additional support, including early cognitive interventions for infants[10 43] and emotional and social support for mothers at the time of and following diagnosis. Furthermore, these mothers may need to be fast-tracked into services, given the particular double vulnerability of HIV and teen pregnancy.

Of note, the overall child sample underperformed somewhat compared with the American normative sample[24] and a similar group of children in Zimbabwe, who had caregivers living with HIV.[44] Although the level of disability did not differ across the three groups, 26% of the total sample scored for any disability. Targeted interventions could improve developmental outcomes,[45] and adapted disability services may be needed for many of these young children. The results from the current study indicate that addressing factors such as young pregnancy or poor maternal education could be of particular benefit. Maternal school progression delay in particular has been shown to be a risk factor for both early motherhood and HIV infection, as well as an outcome of perinatal HIV infection.[46–48] Therefore, it will be important to study this factor as a risk marker for child poor developmental outcomes. However, it is important to note that our model only explained a low amount of variance overall, suggesting that other factors may be key for improving child developmental outcomes.

It is important to note though that all adolescent mothers and their children in the current sample had a number of challenges to contend with. Only 24.6% attended crèche (no difference by group). The mean age for first childbirth was 16.76 years, with the teens living with HIV being on average about a year older. Over 90% were in receipt of some form of household grant. This highlights the compounded disadvantages that adolescent/young mothers living in marginalised areas may face, and the urgent need for support and intervention.[22] In accordance with previous literature,[16] adolescents faced different challenges depending on whether they were perinatally or recently infected (lower social support in the third generation vs food insecurity and mental-health related difficulties in the second generation). This suggests second generation mothers may require additional support in the form of social protection and access to services.[49 50] Mothers of the third generation, on the other hand, appear to have adapted to some degree, but may profit from initiatives to reduce chronic stigma and improve social support in an environment that has been long-term affected by HIV.[51]

The current study has several limitations. First, the data collected are cross-sectional in nature, and thus do not allow any deductions about temporal relations or causality. Furthermore, they stem solely from South Africa and sampling focused predominantly on a particular group (adolescent mothers from disadvantaged environments) and was thus not completely random, potentially limiting the generalisability of our findings to other countries and contexts. Third, the majority of maternal data were collected through self-report, allowing the possibility for bias. Particularly maternal mental health difficulty rates were overall very low in this sample, leading to the question of whether this is due to high resilience or measurement difficulties. Fourth, as the study was community-based, and clinic records were wanting, an algorithm was required to categorise perinatal infection. Fifth, the standardised developmental inventory has been validated and used widely in sub-Africa (eg, Zimbabwe, Zambia, South Africa), but is set against USA norms and local inventories with local references groups may be more sensitive for future studies. Sixth, while our sample predominantly comprised adolescent mothers, some older mothers up to age 25 whom we had historical data for were also included in our analyses to increase analytical power. This limited our ability to draw conclusions for adolescent mothers specifically. Furthermore, our study may not generalise to older mothers and data on fathers could help to further enhance our understanding. Yet it seems important that real life multiple challenges (ie, compounded effects of HIV exposure and young motherhood) should be appreciated, and these insights applied to wider groups. Finally, the group of children of the third generation was relatively small, which—while encouraging giving HIV preventative efforts—limited our power to investigate group differences and to perform more complex analyses, such as mediation or path models, to better understand pathways for intervention. Future studies may aim at specifically recruiting a larger sample of third generation children to replicate and extend our findings. The effects of child HIV on cognitive outcomes in the overall sample are explored elsewhere (Roberts *et al*, in preparation).

## CONCLUSION

The current paper takes an important first look at the children of the 'third generation', giving an impression of their developmental status and life circumstances. The data tentatively suggests that young mothers who have lived in families with HIV for many years learn to understand and accommodate the situation, have worked out pathways to support, and may not have to deal with the shock of HIV diagnosis at the time of pregnancy. Newly infected adolescents may not have the same level of family support and understanding, and they may be coming to terms with their diagnosis at the same time as adjusting to motherhood. For all children, access to early childhood programmes was low, despite them having a good evidence base for enhancing child outcomes. Policies around such provisions should be incorporated for these groups. Future studies will benefit from longitudinal data, including a larger group of children of mothers perinatally infected with HIV, to better understand child cognitive development of the second and third generation, and how this is influenced by maternal factors. While the long-term aim will be to eradicate perinatal and later life HIV infections completely, it is important to further understand how to support affected families adequately.

### Author affiliations
[1]Institute for Global Health, University College London, London, UK
[2]Norwegian Institute for Public Health, Oslo, Norway
[3]Department of Social Policy & Intervention, University of Oxford, Oxford, UK
[4]Department of Psychiatry and Mental Health, University of Cape Town, Rondebosch, Western Cape, South Africa
[5]Department of Infectious Disease Epidemiology, London School of Hygiene & Tropical Medicine, London, UK
[6]Centre for Social Science Research, University of Cape Town, Rondebosch, Western Cape, South Africa
[7]School of Humanities, Sol Plaatje University, Kimberly, South Africa
[8]Institute for Life Course Health Research, Department of Global Health, Faculty of Medicine and Health Sciences, University of Stellenbosch, Stellenbosch, Western Cape, South Africa
[9]Department of Sociology, University of Cape Town, Rondebosch, Western Cape, South Africa

**Acknowledgements** The authors are grateful to the young mothers and their families, the tireless data collection team and partner organisations who supported the research process. We also thank our funders, including FHI360, the European Research Council, US Agency for International Development, USAID, United States President's Emergency Plan for AIDS Relief, Research England, UK Medical Council, UK Department for International Development, National Institute of Health Research, Leverhulme Trust, University College London HelpAge, UNICED Eastern and Southern Africa Office, International AIDS Society, International AIDS society, UKRI GCRF Accelerating Achievement for Africa's Adolescents Hub, Fogarty International Center, National Institute on Mental Health, National Institutes of Health, Oak Foundation and the Economic and Social Research Council.

**Contributors** LDC, ET and LS act as guarantors for the study and conceptualised the overall study. ET, LDC, CW, BS, NL, JJ, JJC-C, MM and WS were involved in study design, data collection and/or data management. KH led and KSR, JT, WS and SZ contributed to the current analyses. LS and KH were predominantly responsible for the writing of the paper, with contributions from the rest of the team. Two adolescent advisory groups served as advisors for contents and acceptability of the questionnaires and recruitment strategies. Community leaders served as advisors for recruitment.

**Funding** This study received funding from the European Research Council (ERC) under the European Union's Horizon 2020 research and innovation programme (grant agreement No 771468); the United States Agency for International Development (USAID) Cooperative Agreement No. AID-OAA-LA-13-00001 and was made possible by the generous support of the American people through USAID and the US President's Emergency Plan for AIDS Relief. The contents are the responsibility of Oxford University and do not necessarily reflect the views of USAID or the United States Government; Research England (0005218); jointly from the UK Medical Research Council (MRC) and the UK Department for International Development (DFID) under the MRC/DFID Concordat agreement, and by the Department of Health Social Care (DHSC) through its National Institutes of Health Research (NIHR) (MR/R022372/1); the Leverhulme Trust (PLP-2014-095); UCL's HelpAge funding; UNICEF Eastern and Southern Africa Office (UNICEF-ESARO); from a CIPHER grant from International AIDS Society (2018/625-TOS). The views expressed in written materials or publications (OR presented in this presentation) do not necessarily reflect the official policies of the International AIDS society; UKRI GCRF Accelerating Achievement for Africa's Adolescents (Accelerate) Hub (Grant Ref: ES/S008101/1); the Fogarty International Center, National Institute on Mental Health, National Institutes of Health under Award Number K43TW011434. The content is solely the responsibility of the authors and does not represent the official views of the National Institutes of Health; Oak Foundation (grant number: OFIL-20-057); the Economic and Social Research Council (grant number ES/R501037/1); the Economic and Social Research Council (grant number ES/J500112/1) the Grand Union - Doctoral Training Partnership; an Economic Social Research Council (ESRC) PhD studentship through the UBEL DTP.

**Competing interests** None declared.

**Patient and public involvement** Patients and/or the public were involved in the design, or conduct, or reporting, or dissemination plans of this research. Refer to the Methods section for further details.

**Patient consent for publication** Not applicable.

**Ethics approval** Ethical approvals were obtained from the Universities of Oxford (R48876/RE002) and Cape Town (HREC 226/2017) as well as University College London for secondary data analysis (14795/001). Additional local approvals and permissions were obtained from the Provincial Departments (Eastern Cape, South Africa) of Health, Education and Social Development. The current research conforms to the principles embodied in the Declaration of Helsinki. Participants gave informed consent to participate in the study before taking part.

**Provenance and peer review** Not commissioned; externally peer reviewed.

**Data availability statement** Data are available on reasonable request. The dataset is available on request.

### ORCID iDs
Lorraine Sherr http://orcid.org/0000-0002-5902-8011
Katharina Haag http://orcid.org/0000-0001-5358-7894
Jenny J Chen-Charles http://orcid.org/0000-0002-5040-0905
Elona Toska http://orcid.org/0000-0002-3800-3173

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
