## [Reviewer comments · BMJ Open]

ARTICLE DETAILS

TITLE (PROVISIONAL)	The development of young children born to young mothers with no, first or second generation HIV acquisition in the Eastern Cape province, South Africa: a cross-sectional study
AUTHORS	Sherr, Lorraine; Haag, Katharina; Roberts, Kathryn; Cluver, Lucie; Wittesaele, Camille; Saliwe, Bongwiwe; Tolmay, Janke; Langwenya, Nontokozo; Jochim, Janina; Saal, Wylene; Zhou, Siyanai; Marlow, Marguerite; Chen-Charles, Jenny; Toska, Elona

VERSION 1 – REVIEW

REVIEWER	Engebretsen, Ingunn Universitetet i Bergen Institutt for indremedisin, Centre for International Health
REVIEW RETURNED	13-Dec-2021

GENERAL COMMENTS	Reviewer comment to "THE CHILDREN OF YOUNG MOTHERS BORN WITH HIV – A study of the cognitive development of infants who are the Third Generation affected by HIV." First and foremost, thank you for letting me review this important paper. The paper covers an important topic and has good data which I think should be published. Having said that, I think the paper could be substantially improved major text revisions, and maybe some reanalysis of certain sections. Major What is generally confusing is the shift in explanation about sampling of mothers to the study on children. E.g. in the abstracts we hear about 1015 young mothers, but the results present findings from 725 + 27 +264children. I think the methods better should reflect the children of interest and that the language throughout the entire manuscript should reflect criteria related to the children (e.g. the abstract is not giving age and sex of children), the reader do not know on which basis children are included. Abstracts The abstract should have its key findings presented as linear regression coefficients with 95% CI not the p-values. That indicates the difference, but also that the sample size (and thereby precision) is what it is. Also, the lack of power is a discussion point and does not belong to the result. It's true that it is not powered, but looking at the numbers they are similar, and that could be emphasized: In other words: It's not interesting to say that these are not significantly different because of power – it's because they are similar (non-infected vs 3rd gen)
--

	Gross motor M: 50.4 vs 52.4; Fine motor: M: 44.4 vs 44.3 and Composite score: M 94.1 vs 94.7. Minor As a person having English as my second language, I cannot comment on the English language that I guess is fine, but I think the authors should pay some attention to typography both in the text and the tables. In the text I find it confusing that not the standard style is used with respect to brackets and full stop. The tables would benefit from being mid- or right aligned (particularly T4). Introduction It is generally a lengthy introduction that needs some streamlining. The concept 'developmental outcomes' is used multiple places in the introduction and elsewhere in the manuscript. When do the authors think about cognitive performance indicators, wealth- and educational outcomes, growth? I think more specifications would be an advantage many places, or at least, a specification of what is meant by it. Second, the objectives are mentioned twice, and I think the introduction needs to be streamlined to present it once in the end of the introduction. Also, The various paragraphs could be more concise and give an introduction to the topic – the various views and then a concluding remark on what the research gap is. The paragraph starting line 38 with 'of note' is a generally well written justification, and could be kept leading up to the objectives. The age group 0-68 is mentioned. Did you really include and test 0 month old? Can you give a justification for the correct age group? I think the introduction would be improved if it is shorter, specified and more structured in general. Other comments from the introduction: Could you get more details about pregnancies under 19 years? (e.g. stratify the younger age groups) The authors may scrutinize the epidemiological literature both the HIC infection and/or the treatment and socio-economic conditions as risk factors for certain developmental outcomes (e.g. early growth (e.g. V. Ramokolo)). Saying "virus and environmental factors" is too rudimentary at this level. Methods The methods would benefit from a more standard structure. It is hard to differentiate the various sections clearly. Could it be structured into Site Design Population and Sampling? Ethics and Data sharing can come after statistical analysis. I wonder if the HEY Baby study is a larger study than the presented cross-sectional study and I also wonder if there are other objectives than what is presented in this paper. The statement on sample size is highly imprecise and does not give the reader any link between Hey Baby and the current study. It is difficult to read information about sampling, cause it is difficult to distinguish whether there are criteria relevant for the mothers or children which is determining the sampling strategies. Also, the age requirement was unclear, I asked about that earlier. And, "on first child" does that mean the adolescent or young mother's first child? What was the rationale for the wide age span of mothers: 10-25 years? The analysis is not well described: This is particularly relevant for the regression analysis. Although it's stated it is hierarchical, T4 is not making it very clear what the logic is going from step 1 to 2.
--	---

	For instance, physical health varies across the groups, but is not visible in any of the steps. Also, what type of modelling technique is used? Results The 1st para mentioned ECD program attendance. Which program is this? Also, the desiccative results are descriptive – why refer to the significance in post-hoc tests (line 31) – that sentence is redundant and can be deleted. Most of the results in T1 are small, except some child variables such as child feeding, and the variability in column percentages speak its own language and do not need to be excessively repeated, neither is the test statistics very interesting. Larger differences between infected and noninfected mothers/children are interesting and can be mentioned in a more condensed manner. The p-values in T1 are not interesting as if statistically significant only are relating to relatively huge differences in the largest population groups. Please delete the column. Could you sort the variables in categorical and continuous? The continuous variables would benefit from having median and range precented too. Also, indicate units for age. I agree that 1 decimal place is enough for the percentages. Please apply that to the continuous variables too. Some of the same challenges as for T1 applies to T2. 1 decimal place is enough. Most of the differences were small except for some indices for the second generation which was somewhat lower. The relevant domains could be mentioned and actually mean differences with 95% CI would have been easier to read. Next, the consideration of 'lack of power' does not belong to the result section. A thorough discussion on limitations considering Type II error related to both sample size and or "sampling" for that sake should be elaborated. Which clinical difference on a continuous score variable is important to detect? Was the sample size based on that? The last finding on children being sick belongs to the beginning of the result section. Should be incorporated in the regression analysis? Page 16 (or 19/28 printed) is lengthy. Could key findings be presented in separate paragraphs and also, please do not present too much test statistics in the text. Further for T4, the 3rd generation consists of 27 people out of more than 1000, almost 2.5%. An external statistician may disagree, but I think it is very interesting to view the descriptive findings by exposure time (across generations), but I do not think it is interesting to look statistical modelling in T4 knowing that a group with 27 is included obviously generating very wide confidence intervals. My suggestion would be to either merge the group with 27 3rd generation children with 2nd generation children or omit them from the model and do linear regression models with HIV infected/exposed vs non-exposed as the main predictor variable. As mentioned earlier, I think it would be better to display a column with crude and adjusted findings and be more transparent with what (does not) go into the final. The key findings from an adjusted model would be the key results from this paper in my view and should be reflected in the abstract.
--	---

	Discussion The discussion would also benefit from editing. The discussion should focus on larger differences and on the findings from the revised modelling. Also, please pay attention to what is introduced which is not explained, e.g. p 21 "only 24.6% attended any early child development programme.= Does this make this study site special – are they public or private or what is it? Public health service or kindergarten or something else? Sampling is special for this study, could that be further elaborated in the limitation section? All the best, Ingunn
--	---

REVIEWER	Njom Nlend, Anne Esther National Insurance Fund Welfare Hospital
REVIEW RETURNED	14-Dec-2021

GENERAL COMMENTS	The paper is good and provide insights around the outcome of 2 schemes of HIV pediatric acquired infection. Children of HIV perinatally infected versus those of mothers behaviourally infected. These are two distinct schemes of acquiring HIV. Many informations may need further investigation in regards to differences observed in motor scales between those children including the mental profile of their mothers.
--

VERSION 1 – AUTHOR RESPONSE

Reviewer 1

Reviewer comment to "THE CHILDREN OF YOUNG MOTHERS BORN WITH HIV – A study of the cognitive development of infants who are the Third Generation affected by HIV."

First and foremost, thank you for letting me review this important paper. The paper covers an important topic and has good data which I think should be published. Having said that, I think the paper could be substantially improved major text revisions, and maybe some reanalysis of certain sections.

We would like to thank the reviewer for the detailed feedback, which we have aimed to address in the following.

Major

What is generally confusing is the shift in explanation about sampling of mothers to the study on children. E.g. in the abstracts we hear about 1015 young mothers, but the results present findings from 725 + 27 + 264 children. I think the methods better should reflect the children of interest and that the language throughout the entire manuscript should reflect criteria related to the children (e.g. the abstract is not giving age and sex of children), the reader do not know on which basis children are included.

We have updated the abstract to state which three groups are being compared in the methods instead of the results section, so the overall sample size and group split are covered in the same place. Information on age and sex of the children is now provided. The introduction section now also focuses on the groups of interest earlier on. Detailed recruitment and inclusion criteria are outlined in the methods section.

Abstracts

The abstract should have its key findings presented as linear regression coefficients with 95% CI not the p-values. That indicates the difference, but also that the sample size (and thereby precision) is what it is.

A sentence summarizing the regression findings has been included in the abstract. We also retained the group means, as they illustrate the (lack of) differences found between the third generation and children of mothers not living with HIV, which is one of the core findings of the paper.

Also, the lack of power is a discussion point and does not belong to the result. It's true that it is not powered, but looking at the numbers they are similar, and that could be emphasized: In other words: It's not interesting to say that these are not significantly different because of power – it's because they are similar (non-infected vs 3rd gen) Gross motor M: 50.4 vs 52.4; Fine motor: M: 44.4 vs 44.3 and Composite score: M 94.1 vs 94.7.

Recent debates surrounding replicability have extensively highlighted power issues in psychological research. Therefore, while we state that the values were approximately equal between the third generation and non-infected groups and discuss the potential implications of these findings, we believe it is also important to highlight to the reader that these conclusions can only be treated as preliminary based on limited power, with further research being required.

Minor

As a person having English as my second language, I cannot comment on the English language that I guess is fine, but I think the authors should pay some attention to typography both in the text and the tables. In the text I find it confusing that not the standard style is used with respect to brackets and full stop. The tables would benefit from being mid- or right aligned (particularly T4).

The manuscript has been thoroughly re-checked for language and typography errors. As far as we are aware, the type of brackets used for references and the associated spacing of full stops reflect the journal's guidelines for reference formatting, but we are happy to update this in case we have misinterpreted them.

Regarding the tables, the values were already mid-aligned as suggested by the reviewer- we are not sure whether the format of the table may have been changed during the submission process, so that this was not visible?

Introduction

It is generally a lengthy introduction that needs some streamlining. The concept 'developmental outcomes' is used multiple places in the introduction and elsewhere in the manuscript. When do the authors think about cognitive performance indicators, wealth- and educational outcomes, growth? I think more specifications would be an advantage many places, or at least, a specification of what is meant by it.

The concept "developmental outcomes" is used as an umbrella term, as we cite studies investigating different domains. Where applicable, we have added details in brackets throughout the introduction section. In terms of our own study, the Mullen scales aim to capture child developmental outcomes in the gross and fine motor, visual reception and expressive and receptive language domains. This has also been added in brackets to the last paragraph of the introduction section.

Second, the objectives are mentioned twice, and I think the introduction needs to be streamlined to present it once in the end of the introduction. Also, The various paragraphs could be more concise and give an introduction to the topic – the various views and then a concluding remark on what the research gap is. The paragraph starting line 38 with 'of note' is a generally well written justification, and could be kept leading up to the objectives.

We have aimed to update and shorten the introduction section accordingly.

The age group 0-68 is mentioned. Did you really include and test 0 month old?

The youngest included child was 2 months old. The paper has now been updated to reflect this.

Can you give a justification for the correct age group? I think the introduction would be improved if it is shortent, specified and more structured in general.

The age groups included were those for whom the Mullen Scales were normed. This has been described in detail in the methods section.

Other comments from the introduction: Could you get more details about pregnancies under 19 years? (e.g. stratify the younger age groups)

We did not stratify our analyses by maternal age originally, as this would have reduced our analytical power even further. However, we have now included sensitivity analyses for the results presented in Table 2, focusing on the adolescent mother sub-sample only (Appendix 1). The direction and strength of differences reflects those in the full sample.

The authors may scrutinize the epidemiological literature both the HIC infection and/or the treatment and socio-economic conditions as risk factors for certain developmental outcomes (e.g. early growth (e.g. V. Ramokolo)). Saying “virus and environmental factors” is too rudimentary at this level.

Some specific factors that were identified through relevant reviews have been added.

Methods

The methods would benefit from a more standard structure. It is hard to differentiate the various sections clearly. Could it be structured into Site Design Population and Sampling? Ethics and Data sharing can come after statistical analysis.

The Ethics statement has been moved to after the statistical analysis section. We have checked the journal guidelines regarding the section headings and have retained the original names, as these are commonly used in our field, and BMJ Open does not suggest any specific headings.

I wonder if the HEY Baby study is a larger study than the presented cross-sectional study and I also wonder if there are other objectives than what is presented in this paper. The statement on sample size is highly imprecise and does not give the reader any link between Hey Baby and the current study.

It is difficult to read information about sampling, cause it is difficult to distinguish whether there are criteria relevant for the mothers or children which is determining the sampling strategies.

The Hey Baby study is a separate study that has a right on its own. Some of the participants were young mothers recruited from a previous study for which we had historical data ($n = 159$: “Mzantsi Wakho”)- this has now been mentioned in the methods section. Additional adolescent mothers were recruited through the channels described in the methods section. A follow up data collection is currently being conducted during the COVID-19 pandemic.

The objectives of the Hey Baby study were to describe 1) the life circumstances of young (predominantly teen) mothers living in adverse circumstances in SSA and 2) the intergenerational effects of HIV, as is mentioned in the introduction section. This has now also been reiterated in the methods section. As shown in the flowchart in Figure 1, the current sample only includes children for whom the MSEL were completed (i.e. who were within the normed age range).

Also, the age requirement was unclear, I asked about that earlier. And, “on first child” does that mean the adolescent or young mother’s first child?

The recruitment focus for Hey Baby was on adolescent mothers. Older mothers are from the “Mzantsi Wakho” study (see above) but were included to increase analytical power. This is now mentioned in the text. The phrasing “first child” relates to “first child of the young mother” – this has now been clarified.

What was the rationale for the wide age span of mothers: 10-25 years?

The age span reflects the range of mothers recruited into the Hey Baby study. While this study predominantly focused on adolescent pregnancy, we decided to keep non-adolescent mothers in the current analyses to not further reduce our already limited power, as outlined above. This has now also been clarified in the “participants” section.

The analysis is not well described: This is particularly relevant for the regression analysis. Although it’s stated it is hierarchical, T4 is not making it very clear what the logic is going from step 1 to 2. For instance, physical health varies across the groups, but is not visible in any of the steps.

As stated in the methods section, the second step of the regression analyses included a range of control variables, in order to explore whether the predictive effects of familial HIV exposure would still hold once these variables are included. Physical health was not included as a predictor, as it is a child-related outcome that was found to differ depending on familial HIV status. It would also be expected to co-vary with physical development scores, as a result of common environmental influences, requiring more complex modelling than possible with the current sample

size. Clarifications about the steps taken in the regression analysis have been added to the heading of table 4.

Also, what type of modelling technique is used?

As outlined in the statistical analysis section, we used univariate ANOVAS to compare the three groups with different familial HIV infection patterns on child development scores and maternal factors, and then conducted hierarchical linear regressions to see if the predictive effects on the MSEL total score would hold once controlling for relevant covariates.

Results

The 1st para mentioned ECD program attendance. Which program is this?

We have changed this to “creche”, as this reflects the type of care received by the children most closely.

Also, the descriptive results are descriptive – why refer to the significance in post-hoc tests (line 31) – that sentence is redundant and can be deleted.

We performed univariate ANOVAS to compare the three groups. As ANOVAs usually do not provide an indication of which groups are significantly different, we additionally included the results from the post-hoc tests.

Most of the results in T1 are small, except some child variables such as child feeding, and the variability in column percentages speak its own language and do not need to be excessively repeated, neither is the test statistics very interesting. Larger differences between infected and noninfected mothers/children are interesting and can be mentioned in a more condensed manner. The p-values in T1 are not interesting as if statistically significant only are relating to relatively huge differences in the largest population groups. Please delete the column.

For consistency and in accordance with journal guidelines, we have decided to retain the values.

Could you sort the variables in categorical and continuous? The continuous variables would benefit from having median and range presented too.

Table 1 is now ordered to first show continuous and then categorical variables for both mother and child. We did not include the median and range as this otherwise would make the tables overly lengthy.

Also, indicate units for age.

This has now been added.

I agree that 1 decimal place is enough for the percentages. Please apply that to the continuous variables too.

This has been changed.

Some of the same challenges as for T1 applies to T2. 1 decimal place is enough. Most of the differences were small except for some indices for the second generation which was somewhat lower. The relevant domains could be mentioned and actually mean differences with 95% CI would have been easier to read.

Decimal places have been adjusted. In the text, we already reported the means and associated post-hoc tests only when ANOVAS were significant. The reason why we did not report the results from the post-hoc tests only is that they may be less illustrative. I.e., only including the difference scores does not convey the point that descriptively, the third generation had mean values that were quite similar to the children of non-infected mothers, which is one of the core findings of interest. For post-hoc tests, the difference scores and associated confidence intervals are now provided instead of the p-values, as suggested by the reviewer.

Next, the consideration of ‘lack of power’ does not belong to the result section. A thorough discussion on limitations considering Type II error related to both sample size and or “sampling” for that sake should be elaborated. Which clinical difference on a continuous score variable is important to detect? Was the sample size based on that?

This was mentioned mainly to remind the reader of the different sample sizes, as it may otherwise be confusing why some of the differences are significant and others not. However, we agree that referring to this several times across the paragraph may have been excessive, and it is now only mentioned once.

The last finding on children being sick belongs to the beginning of the result section. Should be incorporated in the regression analysis?

We treated child health/ illness as a developmental outcome, as it may be influenced by environmental factors such as maternal well-being and ability to care for the child. Thus, it may covary with cognitive and motor development, rather than predicting it (see above).

Page 16 (or 19/28 printed) is lengthy. Could key findings be presented in separate paragraphs and also, please do not present too much test statistics in the text.

As above, we included only statistics for significant differences and the associated post-hoc tests in the text. The findings are now however split into two paragraphs to increase readability.

Further for T4, the 3rd generation consists of 27 people out of more than 1000, almost 2.5%. An external statistician may disagree, but I think it is very interesting to view the descriptive findings by exposure time (across generations), but I do not think it is interesting to look at statistical modelling in T4 knowing that a group with 27 is included obviously generating very wide confidence intervals. My suggestion would be to either merge the group with 27 3rd generation children with 2nd generation children or omit them from the model and do linear regression models with HIV infected/exposed vs non-exposed as the main predictor variable.

We completely agree with the reviewer's comment about limited power and have highlighted this several times throughout the paper. However, the main aim of the current paper was to investigate the development of the third generation. There is already a substantial literature on the cognitive development of children according to maternal HIV status, so collapsing the groups would mean a missed opportunity to gain additional insights into a -so far- under-studied group.

We have aimed to acknowledge limited power by only including a very limited set of control variables when investigating whether predictive effects of familial HIV infection appear to hold, and by consistently discussing the findings as tentative and highlighting a requirement for further research. We have now also added a recommendation to the limitations section that future studies should aim to recruit more individuals from the third generation group specifically, to replicate and extend current findings.

As mentioned earlier, I think it would be better to display a column with crude and adjusted findings and be more transparent with what (does not) go into the final. The key findings from an adjusted model would be the key results from this paper in my view and should be reflected in the abstract. In table 4, step 1, we present the findings of the regression analyses with only familial patterns of HIV as a predictor. In the second step, we adjust for covariates. All of the covariates are kept in the adjusted model, so the model reported in the second step presents adjusted findings. It has now been highlighted in the text and table which steps reflect the crude and the adjusted findings respectively.

Discussion

The discussion would also benefit from editing. The discussion should focus on larger differences and on the findings from the revised modelling. Also, please pay attention to what is introduced which is not explained, e.g. p 21 "only 24.6% attended any early child development programme.= Does this make this study site special – are they public or private or what is it? Public health service or kindergarten or something else?"

We have not focused as strongly on the findings from the regression model in the discussion section, as we found that it only explains a small amount of variance in the MSEL total score, which is mentioned in the results section. Thus, we did not want to over-emphasize any findings made based on this model.

As highlighted above, any phrasing relating to ECD has now been changed to "creche", as this most accurately reflects the type of care the children would have received.

Sampling is special for this study, could that be further elaborated in the limitation section?

As mentioned in response to earlier comments, we have provided additional details on the sampling in the methods section and also included sensitivity analyses focusing on adolescent mothers only in the Appendix. We furthermore added an additional sentence highlighting that the sample consisted predominantly of adolescent mothers, but that some mothers older than 19 years of age were included to increase power into the limitations section.

Overall, however, we believe that the diverse recruitment methods, which were developed in cooperation with adolescent and community advisors, are a particular strength of the study, which to our knowledge is the largest study globally on adolescent mothers living in an environment strongly affected by HIV and deprivation.

Reviewer 2:

Comments to the Author:

The paper is good and provide insights around the outcome of 2 schemes of HIV pediatric acquired infection.

Children of HIV perinatally infected versus those of mothers behaviourally infected.

These are two distinct schemes of acquiring HIV. Many information may need further investigation in regards to differences observed in motor scales between those children including the mental profile of their mothers.

We would like to thank reviewer 2 for their positive feedback and agree that additional research and information is required on this important topic.

1

VERSION 2 – REVIEW

REVIEWER	Engebretsen, Ingunn Universitetet i Bergen Institutt for indremedisin, Centre for International Health
REVIEW RETURNED	26-May-2022

GENERAL COMMENTS	Review: Thanks for letting me review this manuscript again. Re-reading a much clearer document makes me see other things. Sorry for bringing in new things this way. My comments are mostly minor. Please consider my suggestion regarding the title and also about some redundant writing in the background. There were a few places some typographical errors and some places things were hard to read. I've listed that below. Also, kindly be more elaborate in your figure and heading titles. I trust the authors manage these issues easily. First, the title: The word infant is confusing as that term infant is for children < 12 months of age. The first part of the title says children. This makes sense as you are excluding when child is > 68 months, but infants in the next part of the title is confusing. Then you are stressing the "third" generation in the title. I do not think that makes a lot of sense as 726/1046 are never affected. Could something like this work: A study of the cognitive characteristics in young children of young mothers with no, first or second generation HIV acquisition in Easter Cape, South Africa: a cross-sectional study Alt: A study of the cognitive characteristics in young children not being, or being exposed to HIV through second or third generation HIV infected young mothers in Easter Cape, South Africa: a cross-sectional study The trial profile title is not coming through well in the downloaded pdf. The title is missing
--

	Appendix 1: There seems to be something not fitting between the title and the table. The table seems to present frequencies and percentages, but it's not given what is presented above. Is it means with SD? Should be specified and be presented in the title and readable within the table. Abstract: Methods: Different numbers from figure Results: Clear and nice. I prefer if also the B-coeff with 95% CI only use one decimal place (precision levels are not higher just because of regression, applies to manuscript too). Also, kindly provide the prevalence estimates of disabilities before stating there is no difference. Conclusion: I think you could switch 1st and 2nd sentence Background: Ref 7-9 largely refers to own studies. The authors could explore some other studies on child growth and other health outcomes than cognitive outcomes which could shed light on developmental outcomes and risks in children, consider Ramokolo V. and Goga A.'s work. From line 43-56: watch out a bit – the language here is very repetitive and no new information is given to the reader Methods: P11, line 45: Kindly specify on which criterion you considered something associated. This should preferable be a bit 'generous' (not $p < 0.05$) as the sample size is small in some of the groups. Results: P 12, line 31: Rather than putting F and p-values, since you are mentioning a group difference, rather write out the mean difference with 95% CI. Line 36: Delete "in post-hoc tests" Line 38, indicate if both the formula and breastfeeding are exclusive (only), as you did that above Indicate what "M" is an abbreviation for, should be 'mean' – but spell out Some of the differences described in the text is redundant (the differences are small and not interesting (even if statistically significant patterns are seen). I suggest you state boldly what was similar – and do not provide too much details in the text about that. See e.g. mother's age and father's age etc. That is similar and do not need to be mentioned with too much details. Again for the reporting of differences on the Mullen test, indicate if you found the differences small or moderate – irrespective of their significance level 1st sentence page 15: I see it is included and referred to in the discussion, but with limited information. Has it been normed for or standardised in South Africa? If not, this limitation must be mentioned in the methods and discussion. I know Mullen and that it is largely non-verbal, but still, country adaptation and validation would be necessary for comparisons. For example: it is one place they asked "what would you do if it rains" – and the correct response would be to go inside. In our city, with 300 rainy days a year- the correct answer would be very different. If any social or country adaptations were done, what was done?
--	--

	Table 2: Also indicate what you are reporting, means, SD, percentages etc For the result text to the mother's characteristics. I suggest you focus on the key characteristics of the whole group and only mention the variables when the differences are large (e.g physical health). I suggest you do not present test statistics for this in the text, but simply refer to the table. Table 3 is more than comprehensive enough, and the text is hard to read and does not add too much information value. For table 4: List the variables in the adjusted model Discussion: I suggest you start the largest paragraph on page 21 with the part starting in line 42: "it is important to note" – and the rest of the paragraph. Then I suggest you only let one sentence remain with the theory on social support and stigma and potential differences between generations of transmission. This text above line 42 is lengthy and not very informative and could be reduced. It should also have some references. That data are only from South Africa is not a limitation, but there might be a limitation that tools are not developed for South Africa (as you mention further down) and that the sampling was not systematic and/or random. A bit strange to say a scale is used with "good effect" Page 27: Conclusion. I suggest you delete sentence 2 Minor: Typography: full stops and parenthesis P8: Past tense missing in last sentence P9, line 43 and onwards: I suggest the authors check the entire manuscript and make sure presentation of the groups of mothers/babies follow the same sequence throughout the manuscript. Also make sure terms harmonise. P10, line 34: Double parenthesis P10, line 43: There is a semi-colon (;) and I do not see the link between the items listed and the South-African census items, kindly elaborate so it becomes understandable for a reader knowing some, but not all of the South African schemes) P7 line 57, the mother is in singular and 'child' in plural 'children' could it be rewritten to indicate that it is a mother-child pair you were including? So there is no confusion if there is more than one child per woman? Table 1: the total column HIV status is a bit confusing as that's the average of only two categories. Drop it? P 14, line 43: Obs parenthesis P 14, line 45: Delete range of p-values
--	---

VERSION 2 – AUTHOR RESPONSE

Reviewer Feedback

Thanks for letting me review this manuscript again. Re-reading a much clearer document makes me see other things. Sorry for bringing in new things this way. My comments are mostly minor. Please consider my suggestion regarding the title and also about some redundant writing in the background. There were a few places some typographical errors and some places things were hard to read. I've listed that below. Also, kindly be more elaborate in your figure and heading titles. I trust the authors manage these issues easily.

We thank the reviewer for taking the time to re-review our manuscript and the additional suggestions, which we have addressed as outlined below.

First, the title:

The word infant is confusing as that term infant is for children < 12 months of age. The first part of the title says children. This makes sense as you are excluding when child is > 68 months, but infants in the next part of the title is confusing. Then you are stressing the "third" generation in the title. I do not think that makes a lot of sense as 726/1046 are never affected.

Could something like this work:

A study of the cognitive characteristics in young children of young mothers with no, first or second generation HIV acquisition in Easter Cape, South Africa: a cross-sectional study

Alt:

A study of the cognitive characteristics in young children not being, or being exposed to HIV through second or third generation HIV infected young mothers in Easter Cape, South Africa: a cross-sectional study

We have updated our title as follows: *The cognitive characteristics of young children born to young mothers with no, first or second generation HIV acquisition in the Eastern Cape province, South Africa: a cross-sectional study.*

The trial profile title is not coming through well in the downloaded pdf. The title is missing

The title for Figure 1 (flow-chart showing inclusion into the study and how mothers and children were grouped) has been provided at the end of the manuscript, as per journal guidelines.

Appendix 1: There seems to be something not fitting between the title and the table. The table seems to present frequencies and percentages, but it's not given what is presented above. Is it means with SD? Should be specified and be presented in the title and readable within the table.

Appendix 1 presents both, means/SD and N/%, depending on the outcome of interest. We have now added what we are referring to in the first columns for each outcome, in order to make this clearer.

Abstract:

Methods: Different numbers from figure.

Numbers in the figure have been updated to reflect the correct numbers.

Results: Clear and nice. I prefer if also the B-coeff with 95% CI only use one decimal place (precision levels are not higher just because of regression, applies to manuscript too).

This has been changed.

Also, kindly provide the prevalence estimates of disabilities before stating there is no difference. "

This has been added.

Conclusion: I think you could switch 1st and 2nd sentence

This has been switched.

Background:

Ref 7-9 largely refers to own studies. The authors could explore some other studies on child growth and other health outcomes than cognitive outcomes which could shed light on developmental outcomes and risks in children, consider Ramokolo V. and Goga A.'s work.

Of references 7-11, which are provided in this paragraph, two involve a single author from the current paper, and these are review studies, so relevant summaries of previous work by other authors. We have looked at the works of the two authors suggested by the reviewer in the previous iteration of the comments and have found their work to be too distally related to the current topic. However, in the previous iteration, we added some additional references that were relevant, as suggested by the reviewer.

From line 43-56: watch out a bit – the language here is very repetitive and no new information is given to the reader

Lines 43-56 have been added in order to contextualize our sample and provide a clear description as to why it is unique. Specifically, this paragraph introduces young motherhood as a key characteristic of the sample, as well as its clustering with increased risk for HIV in disadvantaged communities, which we believe is a key consideration and poses new, relevant information for the reader.

Methods:

P11, line 45: Kindly specify on which criterion you considered something associated. This should preferable be a bit 'generous' (not $p < 0.05$) as the sample size is small in some of the groups.

This has been added to the statistical analysis section.

Results:

P 12, line 31: Rather than putting F and p-values, since you are mentioning a group difference, rather write out the mean difference with 95% CI.

This has been added.

Line 36: Delete "in post-hoc tests"

This has been deleted.

Line 38, indicate if both the formula and breastfeeding are exclusive (only), as you did that above.

This has been added.

Indicate what "M" is an abbreviation for, should be 'mean' – but spell out.

The use of *M* for mean is common in our field and does not commonly require a specific definition (see APA guidelines). We are happy to adjust this if required by journal guidelines.

Some of the differences described in the text is redundant (the differences are small and not interesting (even if statistically significant patterns are seen). I suggest you state boldly what was similar – and do not provide too much details in the text about that.

See e.g. mother's age and father's age etc. That is similar and do not need to be mentioned with too much details.

Details were reported because ages of mothers in all three groups were significantly different from each other, while only the second-generation fathers' ages were significantly different from the other two groups. We believe that it is important to provide detailed age information, since young mothers are a particularly vulnerable population, and age is an important factor in relation to possible policy programming and reach.

Again for the reporting of differences on the Mullen test, indicate if you found the differences small or moderate – irrespective of their significance level

We are not entirely sure what criteria the reviewer wanted us to apply in order to determine small versus moderate effects. As per the reviewer's suggestion in the previous iteration, we have included difference scores between groups and associated confidence intervals, which we think is an appropriate way to capture any relevant effects, and the confidence the reader may have in these estimates.

1st sentence page 15: I see it is included and referred to in the discussion, but with limited information. Has it been normed for or standardised in South Africa? If not, this limitation must be mentioned in the methods and discussion. I know Mullen and that it is largely non-verbal, but still, country adaptation and validation would be necessary for comparisons.

For example: it is one place they asked "what would you do if it rains" – and the correct response would be to go inside. In our city, with 300 rainy days a year- the correct answer would be very different. If any social or country adaptations were done, what was done?

The scales have been validated for use in Sub-Saharan Africa but have not specifically normed or adapted. We agree that this is a limitation, and therefore have highlighted in the methods section that the scales have only been normed in the US, and the lack of a normative Sub-Saharan African sample is also critically reflected upon in the discussion section.

Table 2: Also indicate what you are reporting, means, SD, percentages etc

This has been added for every outcome.

For the result text to the mother's characteristics. I suggest you focus on the key characteristics of the whole group and only mention the variables when the differences are large (e.g physical health). I suggest you do not present test statistics for this in the text, but simply refer to the table. Table 3 is more than comprehensive enough, and the text is hard to read and does not add too much information value.

Our paper is largely descriptive, as its main aim is to provide an initial idea of the third generation and their living circumstances, which to our knowledge has not been reported on in the scientific literature to date. Therefore, we report on key differences not only in the table but also in the text. As per the reviewer's recommendation in the previous iteration, we already only briefly report non-significant differences, as this information is available in the table.

For table 4: List the variables in the adjusted model

Variables with non -significant effects are now mentioned in the text.

Discussion:

I suggest you start the largest paragraph on page 21 with the part starting in line 42: “it is important to note” – and the rest of the paragraph. Then I suggest you only let one sentence remain with the theory on social support and stigma and potential differences between generations of transmission. This text above line 42 is lengthy and not very informative and could be reduced. It should also have some references.

We have restructured and shortened the paragraph accordingly.

That data are only from South Africa is not a limitation, but there might be a limitation that tools are not developed for South Africa (as you mention further down) and that the sampling was not systematic and/or random.

A comment about sampling has been added. The lack of a South African normative sample had already been addressed in the text, as the reviewer pointed out.

A bit strange to say a scale is used with “good effect”

The sentence has been restructured to avoid this phrasing.

Page 27: Conclusion. I suggest you delete sentence 2

This has been deleted.

Minor:

Typography: full stops and parenthesis

This has already been addressed in the previous iteration of comments- the full stops and parenthesis reflect the formatting required by the journal for references.

P8: Past tense missing in last sentence

This has been adjusted.

P9, line 43 and onwards: I suggest the authors check the entire manuscript and make sure presentation of the groups of mothers/babies follow the same sequence throughout the manuscript. Also make sure terms harmonise.

This has been checked and it has been ensured that the same terms have been used throughout the manuscript. In terms of reporting the results, since we have restricted ourselves to only reporting significant differences, this has been guided by where such differences have been found.

P10, line 34: Double parenthesis

This has been deleted.

P10, line 43: There is a semi-colon (;) and I do not see the link between the items listed and the South-African census items, kindly elaborate so it becomes understandable for a reader knowing some, but not all of the South African schemes)

The reference is for the specific item that has been used, which is from the South African census (rather than another validated scale). “Measured via” has now been added into the text to make this clearer.

P7 line 57, the mother is in singular and 'child' in plural 'children' could it be rewritten to indicate that it is a mother-child pair you were including? So there is no confusion if there is more than one child per woman?

"Children" has been changed to "child" to reflect this.

Table 1: the total column HIV status is a bit confusing as that's the average of only two categories. Drop it?

We added this number as it may otherwise be confusing for the reader to have one empty cell in the summary table. Given that we have indicated that there is no data for the group not living with HIV using a dash, we hope that the reader will be able to deduct that the total number has been calculated for the other two cells.

P 14, line 43: Obs parenthesis

This has been changed.

P 14, line 45: Delete range of p-values

This has been changed.

VERSION 3 – REVIEW

REVIEWER	Engebreetsen, Ingunn Universitetet i Bergen Institutt for indremedisin, Centre for International Health
REVIEW RETURNED	12-Sep-2022

GENERAL COMMENTS	General comment: Hi authors and editors, I generally find this an informative, important, relevant and good study and in my understanding closer to a final version now. Generally, this is now easier to read and the results are overall well presented. The introduction, methods and discussion parts are generally clear. Most of my prior comments have been adhered to. Sometimes the authors express a wish to write it their way for example with pretty lengthy rewriting of their table results. That's ok with me, I acknowledge some different writing traditions and leave that to the editor to decide. Still, while reading I come across a few more things which I think can be improved before publication which I fully believe the authors manage well without further review. Also, there may be things I did not see last time or something I did not see before, sorry if that's the case. Major comment: My only major comment is that it must be crystal clear at the end of the background where objectives are mentioned and in the beginning of methods what the target age is. Some various statements just increase confusion now. Various age groups are mentioned in the
--

	background and methods, and it would be nice if that is coherently spelled out, preferably once. Minor: Abstract: Abstract, methods: Please rewrite 1st sentence with a subject and object. Please mention the “generations” following the none, ‘second’ and ‘third’ logic flow in the manuscript. Please spell out: Rather than univariate ANOVAs “say “univariate approach, analysis of variance (ANOVA) analysis” Same in results: Spell out hierarchical regressions to “hierarchical regression analyses” Separate strengths and limitations section: Last sentence: Write in plural: ‘samples’ (we want more studies on this topic, not just one) Background: Line 27: Delete word: ‘adolescent’ , just write: “ “third generation”, i.e., children who have their mother perinatally infected by HIV.” Line 38: Spell out ART first time Line 43-45: Do not repeat definition of 3rd generation I suggest you delete sentence: “To our knowledge, no studies have yet investigated the effects of intergenerational HIV infection on child outcomes independent of child HIV status, and also in comparison to a group of children without familial HIV exposure, but who live in the same high-risk environment.” You have already introduced this point in the beginning of the paragraph and argued well for it. Line 17 (p 2 of introduction): you write: “have been proposed to more likely be members of marginalized groups.” Kindly simplify grammar here, e.g, those recently infected have more likely been marginalised and struggle with recent impact of their diagnosis. Point: I do not think marginalised groups necessarily have ‘members,’ I think people are more or less marginalised.... The following sentence can go out of introduction as its key point is mentioned in the method: “The current sample, which predominantly consists of adolescent mothers (age 19 or under) thus provides an unique opportunity to study the compounded effects of HIV and young pregnancy.” Introduction page 3 line 8, write: “in children of mothers 19 years or younger” (if that is correct according to my major comment) Last paragraph, again – I suggest you keep the order of non, 2nd and 3rd generation when presenting. I also suggest you delete the last part of the last sentence: “all of which can shape the environment a child grows up in.” This is nicely introduced in the background and doesn’t need repetition when stating the objectives. Generally the introduction is clear and thus deleting what may be ‘redundant’ may make the well presented messages come more well through.
--	--

	Methods: Line 57: Here you say 'young mother' specify with age what you mean. It's also confusing that you here say sensitivity analysis for younger/equal to 19 as that is what you said in the introduction was special about this study. Please be extremely coherent between your objectives and methods. Page 7, line 57: 'childof' to ' child of' Results: Line 9-10: Kindly delete: 'likely due to maternal age differences (see below)' For age differences, kindly only provide 1 decimal For the paragraph on mothers line 27: write 'were slightly older' (the age difference is the interesting point, not the statistical significance) It is generally no need to provide statistical testing statistics information in the text where you present the baseline characteristics as that is already fully explained in Table 1. I see you argue for doing that in the response to us (me), I guess that's a preference difference and we can leave it like that. But then, when you provide test statistics, kindly avoid 'significant' when you write 'significantly older... etc, just write that they were older/younger, similar etc. I think you should report on the similar creche attendance across the groups, also as you are referring to that in the discussion. To me this is an important characteristic, even that it's similar across the groups. I miss information about parity. Do you have that? The tables are generally tidy, structured and easy to follow now. T1: Avoid two decimals for maternal age, column 2 Page 16, line 22, space is missing after comma I think the Table 3 "Poverty and Social Protection" variables should be reported in table 1 and not in table 3. I think table 3 and text would give a clearer somatic and mental health picture and the text would also be more condensed. Text page 20, kindly avoid giving the p-values in the text, it's ok the way they are in the tables. If you have to give numbers in the text, kindly give the regression coefficients with the 95% CI Are the numbers you give for the step 2 analysis wrong in the text or table? They are not similar. Also, spell out what B and beta is, in the table 4 heading. Ethics: The ethical statement should explain if there were ethical difficulties and if women/babies in need got extra follow-up/referral and how that was arranged. To the prior review: Regarding the title: I think it is better now, but it stroke me that maybe you should stick close to your aim and say 'child development of young children born to...." as you do more than cognitive testing.
--	---

VERSION 3 – AUTHOR RESPONSE

Response to reviewers

General comment:

Hi authors and editors,

I generally find this an informative, important, relevant and good study and in my understanding closer to a final version now. Generally, this is now easier to read and the results are overall well presented. The introduction, methods and discussion parts are generally clear. Most of my prior comments have been adhered to. Sometimes the authors express a wish to write it their way for example with pretty lengthy rewriting of their table results. That's ok with me, I acknowledge some different writing traditions and leave that to the editor to decide. Still, while reading I come across a few more things which I think can be improved before publication which I fully believe the authors manage well without further review. Also, there may be things I did not see last time or something I did not see before, sorry if that's the case.

Thank you for your comments and for taking the time to review.

Major comment: My only major comment is that it must be crystal clear at the end of the background where objectives are mentioned and in the beginning of methods what the target age is. Some various statements just increase confusion now. Various age groups are mentioned in the background and methods, and it would be nice if that is coherently spelled out, preferably once.

Thank you for your comment. The age range of the children who are the focus of this study has now been added to the back ground section of this manuscript and this detail is also within the participants section of the methods.

Minor:

Abstract:

Abstract, methods: Please rewrite 1st sentence with a subject and object.

Please mention the "generations" following the none, 'second' and 'third' logic flow in the manuscript.

Please spell out: Rather than univariate ANOVAs "say "univariate approach, analysis of variance (ANOVA) analysis"

Same in results: Spell out hierarchical regressions to "hierarchical regression analyses"

Thank you these abstracts have now been addressed in the abstract.

Separate strengths and limitations section:

Last sentence: Write in plural: 'samples' (we want more studies on this topic, not just one)

Addressed – thank you

Background:

Line 27: Delete word: 'adolescent' , just write: " "third generation", i.e., children who have their mother perinatally infected by HIV."

Line 38: Spell out ART first time

Line 43-45: Do not repeat definition of 3rd generation

Thank you – these comments have been addressed

I suggest you delete sentence: "To our knowledge, no studies have yet investigated the effects of intergenerational HIV infection on child outcomes independent of child HIV status, and also in comparison to a group of children without familial HIV exposure, but who live in the same high-risk environment." You have already introduced this point in the beginning of the paragraph and argued well for it.

This has been removed

Line 17 (p 2 of introduction): you write: "have been proposed to more likely be members of marginalized groups." Kindly simplify grammar here, e.g, those recently infected have more likely been marginalised and struggle with recent impact of their diagnosis.

Point: I do not think marginalised groups necessarily have 'members,' I think people are more or less marginalised....

This comment has now been addressed in the introduction

The following sentence can go out of introduction as its key point is mentioned in the method: "The current sample, which predominantly consists of adolescent mothers (age 19 or under) thus provides an unique opportunity to study the compounded effects of HIV and young pregnancy."

Introduction page 3 line 8, write: "in children of mothers 19 years or younger" (if that is correct according to my major comment)

Last paragraph, again – I suggest you keep the order of non, 2nd and 3rd generation when presenting.

I also suggest you delete the last part of the last sentence: "all of which can shape the environment a child grows up in." This is nicely introduced in the background and doesn't need repetition when stating the objectives.

Generally the introduction is clear and thus deleting what may be 'redundant' may make the well presented messages come more well through.

Thank you again for your comments these have now been addressed throughout the introduction.

Methods:

Line 57: Here you say 'young mother' specify with age what you mean.

This detail has been added

It's also confusing that you here say sensitivity analysis for younger/equal to 19 as that is what you said in the introduction was special about this study. Please be extremely coherent between your objectives and methods.

This has been amended for clarity

Page 7, line 57: 'childof' to ' child of'

This has been amended

Results:

Line 9-10: Kindly delete: 'likely due to maternal age differences (see below)'

For age differences, kindly only provide 1 decimal

These amendments have been made

For the paragraph on mothers line 27: write 'were slightly older' (the age difference is the interesting point, not the statistical significance)

This change has been made in text

It is generally no need to provide statistical testing statistics information in the text where you present the baseline characteristics as that is already fully explained in Table 1. I see you argue for doing that in the response to us (me), I guess that's a preference difference and we can leave it like that.

We have chosen to keep this as is, based on our previous response.

But then, when you provide test statistics, kindly avoid 'significant' when you write 'significantly older... etc, just write that they were older/younger, similar etc.

This alternation has been made in text

I think you should report on the similar creche attendance across the groups, also as you are referring to that in the discussion. To me this is an important characteristic, even that it's similar across the groups.

This information is included within the first paragraph of the results focusing on children.

I miss information about parity. Do you have that?

The authors decided not to include this information.

The tables are generally tidy, structured and easy to follow now.

T1: Avoid two decimals for maternal age, column 2

This has been amended

Page 16, line 22, space is missing after comma

Amended

I think the Table 3 "Poverty and Social Protection" variables should be reported in table 1 and not in table 3. I think table 3 and text would give a clearer somatic and mental health picture and the text would also be more condensed.

These changes have now been made

Text page 20, kindly avoid giving the p-values in the text, it's ok the way they are in the tables. If you have to give numbers in the text, kindly give the regression coefficients with the 95% CI Where appropriate p- values have been removed. P-values have been retained for posthoc testing. Are the numbers you give for the step 2 analysis wrong in the text or table? They are not similar. Step 2 only includes a significance value and this has been fully checked.

Also, spell out what B and beta is, in the table 4 heading.

This detail has been added

Ethics: The ethical statement should explain if there were ethical difficulties and if women/babies in need got extra follow-up/referral and how that was arranged.

This information has now been added

To the prior review:

Regarding the title:

I think it is better now, but it stroke me that maybe you should stick close to your aim and say 'child development of young children born to....' as you do more than cognitive testing

Thank you for your suggestion. An alteration has now been made to the title